# Comparison of different regimens with or without fluoroquinolone in isoniazid-resistant tuberculosis: A multicenter cohort study

Jinsoo Min[1], Hyung Woo Kim[2], Ji Young Kang[3], Sung Kyoung Kim[4], Jin Woo Kim[5], Yong Hyun Kim[6], Hyoung Kyu Yoon[7], Sang Haak Lee[8], Ju Sang Kim[2]*

1 Division of Pulmonary and Critical Care Medicine, Department of Internal Medicine, Seoul St. Mary's Hospital, College of Medicine, The Catholic University of Korea, Seoul, Republic of Korea, 2 Division of Pulmonary and Critical Care Medicine, Department of Internal Medicine, Incheon St. Mary's Hospital, College of Medicine, The Catholic University of Korea, Seoul, Republic of Korea, 3 Department of Internal Medicine, Cheju Halla General Hospital, Jeju-do, Republic of Korea, 4 Division of Pulmonology, Department of Internal Medicine, St. Vincent's Hospital, College of Medicine, The Catholic University of Korea, Seoul, Republic of Korea, 5 Division of Pulmonary and Critical Care Medicine, Department of Internal Medicine, Uijeongbu St. Mary's Hospital, College of Medicine, The Catholic University of Korea, Seoul, Republic of Korea, 6 Division of Pulmonary, Allergy and Critical Care Medicine, Department of Internal Medicine, Bucheon St. Mary's Hospital, College of Medicine, The Catholic University of Korea, Seoul, Republic of Korea, 7 Division of Pulmonary and Critical Care Medicine, Department of Internal Medicine, Yeouido St. Mary's Hospital, College of Medicine, The Catholic University of Korea, Seoul, Republic of Korea, 8 Division of Pulmonary, Critical Care and Sleep Medicine, Department of Internal Medicine, Eunpyeong St. Mary's Hospital, College of Medicine, The Catholic University of Korea, Seoul, Republic of Korea

* kimjusang@catholic.ac.kr

**Data Availability Statement:** All relevant data are within the manuscript and its Supporting Information files.

## Abstract

In 2018, the World Health Organization recommended a 6-month four-drug regimen (rifampicin, ethambutol, pyrazinamide, and levofloxacin) for the treatment of isoniazid-monoresistant tuberculosis. However, the regimen had very low certainty. This cohort study assessed the impact of fluoroquinolone use and initial baseline regimen on treatment effectiveness in isoniazid-monoresistant tuberculosis. This multicenter retrospective cohort study included 318 patients with isoniazid-monoresistant tuberculosis notified between 2011 and 2018 in Korea. Baseline regimens were classified into two groups, namely 6–9-month rifampicin, ethambutol, and pyrazinamide (6-9REZ) and a combination regimen of 2-month rifampicin, ethambutol, pyrazinamide and 7–10-month rifampicin and ethambutol (2REZ/7-10RE). Multivariable logistic regression was performed to assess factors associated with positive treatment outcomes. Of 318 enrolled patients, 234 (73.6%) were treated with the 6-9REZ and 103 (32.4%) with additional fluoroquinolone. In a multivariable logistic regression model comparing the 6-9REZ and 2REZ/7-10RE groups, there was no difference in the odds of positive outcomes (adjusted odds ratio = 1.08, 95% confidence interval = 0.65–1.82). Addition use of fluoroquinolone was not associated with positive treatment outcomes in the whole cohort (adjusted odds ratio = 1.41, 95% confidence interval = 0.87–2.27); however, its additional use was beneficial in the 2REZ/7-10RE subgroup (adjusted odds ratio = 3.58, 95% confidence interval = 1.32–9.75). Both initial baseline regimens, 6-9REZ and 2REZ/7-10RE, were similarly effective. Shortening of the pyrazinamide administration

**Funding:** This work was supported by the Research Program funded by the Korea National Institute of Health [grant number 2019E520202]. The funder had no role in the study design, data collection and analysis, decision to publish, or preparation of the manuscript. https://www.nih.go.kr

**Competing interests:** The authors have declared that no competing interests exist.

duration with additional fluoroquinolone use could be a safe alternative for patients with potential hepatotoxicity related to pyrazinamide.

## Introduction

Isoniazid (INH, H) is one of the most important first-line medicines for the treatment of active tuberculosis (TB) and latent TB infection, with high bactericidal activity and a good safety profile. Recently, the global prevalence of isoniazid-resistant rifampicin-susceptible TB (Hr-TB) was estimated to be 7.4% among new TB patients and 11.4% among previously treated TB patients, particularly in the European and Western Pacific regions [1]. Hr-TB is much more common than rifampicin resistance and could seriously jeopardize progress in the fight against TB [2].

INH resistance has been associated with poor treatment outcomes, with the possible acquisition of additional anti-TB drug resistance [3,4]. Despite the frequent occurrence of Hr-TB, little research has been conducted to optimize its regimen recommendations. In 2018, the World Health Organization (WHO) guidelines for the treatment of Hr-TB recommended a 6-month four-drug regimen consisting of rifampicin (RIF, R), ethambutol (E), pyrazinamide (PZA, Z), and levofloxacin [5]. However, the WHO assessed the regimen as having very low certainty.

The Republic of Korea has the highest TB incidence among high-income countries. Among native TB patients, the proportions of Hr-TB in new and retreated cases were 6.9% and 8.5%, respectively [6]. Although fluoroquinolones (Fqs) are widely and commonly used for anti-TB treatment under universal health coverage in Korea, their roles in Hr-TB treatment have not been evaluated. We aimed to identify treatment regimens for Hr-TB and assessed the impact of initial baseline regimen choices and additional Fq use on treatment outcomes through a multicenter retrospective cohort study.

## Materials and methods

### Study setting and participants

We conducted a multicenter retrospective cohort study of Hr-TB cases at eight university-affiliated hospitals in the Seoul metropolitan area and Daejeon, Korea. These hospitals participated in the national public-private mix TB control project and provided comprehensive patient management [7]. We included notified patients with pulmonary TB aged ≥15 years between January 2011 and December 2018 in the study cohort. Patient data were collected from electronic medical records. Other inclusion criteria were as follows: (1) patients who had a positive acid-fast bacillus culture test result; (2) patients phenotypically or genotypically confirmed with INH resistance; and (3) patients who started the initial standard four-drug anti-TB treatment regimen of HREZ. Exclusion criteria were as follows: (1) patients who were transferred to other TB clinics before starting anti-TB treatment; (2) patients who died before TB diagnosis; and (3) patients with only extrapulmonary TB. We also excluded patients who initially received HREZ but had their regimen changed within 2 months because it was difficult to evaluate the efficacy of Fq use during the initial 2 months after treatment commencement.

## Treatment outcomes

Treatment outcomes were defined according to the WHO's definition. "Treatment success" was defined as treatment completed as initially prescribed once INH resistance was known, without extending duration of specified regimens. "Positive treatment outcome" was defined as success of treatment without recurrence within the 1-year post-treatment follow-up period. "Unfavorable outcome" was defined as a composite outcome that includes death, treatment failure, loss-to-follow-up, transfer-out, and recurrence.

Recently, the WHO convened a consultation meeting to update treatment outcome definitions [8], in which they proposed a new definition of "treatment failure"—when a treatment regimen is terminated or permanently changed to a new treatment or treatment strategy. Because one of our study objectives was to evaluate the efficacy and toxicity of treatment regimens for Hr-TB, we adopted concept of regimen-specific outcomes [9,10] and defined treatment failure as follows: (1) regimen changed due to adverse events; (2) regimen strengthened due to worsening or not improving; and (3) extended length of treatment due to lack of clinical response.

"No response to treatment" is one of the most important reasons for treatment failure. Therefore, when treatment duration was extended due to lack of clinical response, we defined it as a treatment failure according to the revised WHO's outcome definitions. The 2008 WHO's guideline of drug-resistant TB recommends prescription of the REZ regimen for 6–9 months; we defined a 6-9REZ regimen of more than 10 months as "extended length of treatment." In the United Kingdom, the 2016 National Institute for Health and Care Excellence recommends a 9-month regimen comprising 2 months of REZ, followed by 7 months of RE. This can be extended to 12 months if disease is extensive. We adopted this guideline and considered those prescribed a 2REZ/7-10RE regimen of more than 13 months as having "extended length of treatment."

## Treatment regimens

Drug names, start and end dates, and dates of missed doses were collected to describe the Hr-TB regimens. We counted all the prescribed doses of anti-TB drugs from the first day of anti-TB treatment for each patient. First, baseline regimens were categorized according to the duration of PZA use as follows: ≥90 days for 6-9REZ or < 90 days for 2REZ/7-10RE (Fig 1). Although the TB guidelines recommend initial prescription of standard regimens of HREZ with 2-month use of PZA, duration of PZA use may vary depending on clinical judgements and may be extended due to accidental addition of anti-TB drugs. There were a few cases of prolonged PZA use of >60 days but <90 days without any comments or evidence of adverse events, so we inferred that the physician's first intention was to administer PZA for 60 days, according to the guideline. Therefore, we set the duration of PZA use to 90 days and classified it as 6-9REZ if PZA was administered for >90 days. Second, additional Fq uses were identified and further categorized as baseline regimens. All drugs were administered daily according to Korean guidelines.

## Independent variables

Phenotypic drug susceptibility tests (DSTs) were conducted at the supranational and commercial reference laboratories, which share the same standard operation protocol. The drug susceptibility of *Mycobacterium tuberculosis* isolates was determined using the absolute concentration method with Lowenstein-Jensen medium, as recommended by the WHO. For example, the critical concentration for high and low INH resistance was 1.0 and 0.2 μg/mL,

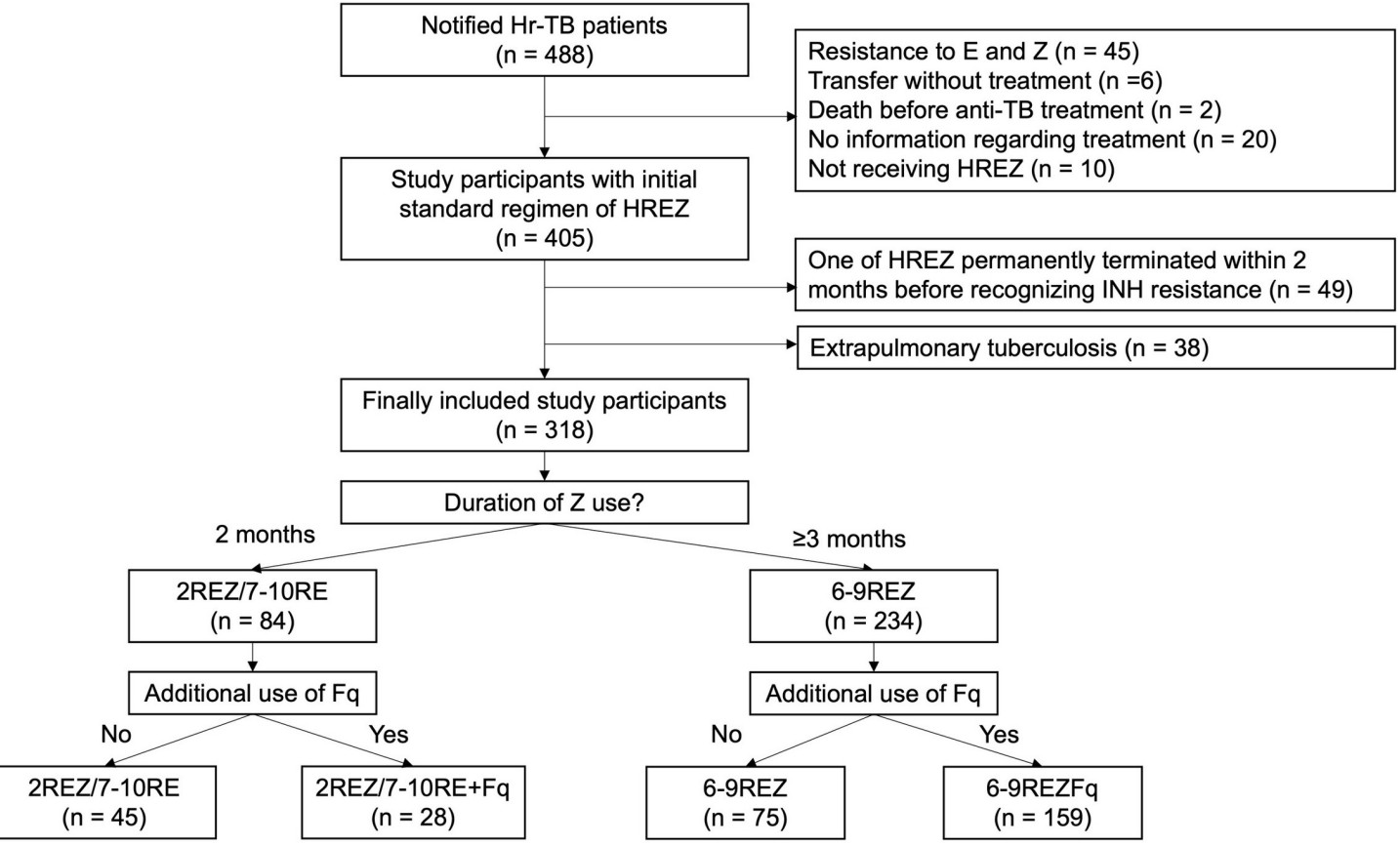

**Fig 1. Flow chart of study participant enrollment.** Hr, isoniazid-resistant; TB, tuberculosis; H, isoniazid; R, rifampicin; E, ethambutol; Z, pyrazinamide; Fq, fluoroquinolone.

respectively. Rapid molecular DSTs were performed using a line probe assay to detect genetic mutations associated with INH resistance.

Data regarding age, sex, nationality, comorbidity, and history of TB treatment were collected. All data were coded as binary data, except for age. The site of TB involvement was combined with acid-fast bacilli smear test results to generate a single variable with three strata, namely pulmonary site with positive smear result, pulmonary site with negative smear result or smear status missing, and extrapulmonary sites without pulmonary involvement.

## Statistical analyses

Continuous variables were presented as means and standard deviations or medians and interquartile ranges (IQRs), whereas discrete variables were presented as frequencies or percentages. The baseline characteristics of enrolled patients with or without additional Fq use were compared; univariable analysis was performed using the chi-square test for categorical variables and the Mann–Whitney U test for continuous variables. Statistical significance was set at $P < 0.05$. All statistical analyses were performed using SPSS version 17.0 (Statistical Product and Service Solutions, Chicago, IL, USA).

For regression analysis, we performed a univariable logistic regression analysis to examine the impact of different factors on the likelihood of positive treatment outcomes. Subsequently, we selected age, sex, baseline regimen, additional Fq use, and other variables with $P < 0.20$

based on the univariable analysis and further constructed a multivariable logistic regression model. We conducted a subgroup analysis to assess the association between additional Fq use and positive treatment outcomes in the different baseline regimens, 2REZ/7-10RE and 6-9REZ.

## Study size

We selected ten variables a priori for inclusion in the model. One hundred events of positive treatment outcomes are required to ensure a minimum of 10 events per variable, which are needed to minimize bias in logistic regression models. Based on a recent Korean study of Hr-TB [11], which demonstrated approximately 84% of treatment success without recurrence duration, at least 120 patients with Hr-TB were required for sample size.

## Ethics statement

This study was conducted in accordance with the principles of the Declaration of Helsinki. The Institutional Review Board (IRB) of the Catholic Medical Center, the Catholic University of Korea approved the study protocol (XC19REDE0040) and waived the need for informed consent because no patients were at risk.

## Results

Among the 488 notified patients with Hr-TB between 2011 and 2018, 405 patients initially started with the first-line anti-TB treatment of HREZ. After excluding 49 patients who had their initial treatment regimen changed within the first 2 months and 38 patients with extra-pulmonary tuberculosis, 318 patients were finally enrolled for analysis. Of these patients, 234 (73.6%) were prescribed 6-9REZ as an initial baseline treatment regimen, and 103 (32.4%) had additional Fq for treatment. Percentage of 6-9REZ as the initial baseline regimen was doubled from 2011–2014 to 2015–2018. Other baseline characteristics of enrolled patients between 6-9REZ and 2REZ/7-10RE were compared and did not reveal any statistically significant differences (Table 1).

There were 160 (50.3%) cases of treatment success without recurrence and 115 (36.1%) cases of treatment failure (Table 2). Among 116 unfavorable outcome cases of the 6-9REZ group, extended length of treatment (44/116, 37.9%) was the most frequent, followed by regimen changed due to adverse events (36/116, 31.0%). However, among 42 unfavorable outcome cases of the 2REZ/7-10RE group, loss-to-follow-up (15/42, 35.7%) was the most frequent, followed by extended length of treatment (8/42 cases, 19.0%).

Percentages of positive treatment outcomes were not significantly different between the 6-9REZ and 2REZ/7-10RE (OR = 1.02, 95% confidence interval [CI] = 0.62–1.68) (Table 3). The addition of Fq to the baseline regimen had more positive outcomes than the baseline regimen without additional Fq use; however, differences in its odds were not significant (OR = 1.35, 95% CI = 0.84–2.16). Except age, none of the other variables was significantly associated with positive outcomes.

In the multivariable model, differences in the odds of positive treatment outcomes were not detected between the different initial baseline regimens (adjusted odds ratio [aOR] = 1.08, 95% CI = 0.65–1.82), and the association between additional Fq use and positive outcomes was insignificant (aOR = 1.41, 95% CI = 0.87–2.27) (Table 4). However, in the subgroup analysis, the magnitudes of association between additional Fq use and positive outcomes were different between 2REZ/7-10RE (aOR = 3.58, 95% CI = 1.32–9.75) and 6-9REZ (aOR = 1.03, 95% = 0.59–1.81) (Fig 2).

**Table 1. Baseline characteristics of 318 enrolled pulmonary tuberculosis patients with isoniazid-resistant rifampicin-susceptible strain stratified by different types of initial baseline regimens.**

| | 2REZ/7-10RE | 6-9REZ | Total | P value |
|---|---|---|---|---|
| | (n = 84) | (n = 234) | (n = 318) | |
| Calendar year | | | | |
| 2011–2014 | 39 (46.4%) | 74 (31.6%) | 113 (35.5%) | 0.015 |
| 2015–2018 | 45 (53.6%) | 160 (68.4%) | 205 (64.5%) | |
| Sex | | | | |
| Female | 40 (47.6%) | 83 (35.5%) | 123 (38.7%) | 0.050 |
| Male | 44 (52.4%) | 151 (64.5%) | 195 (61.3%) | |
| Age, years | | | | |
| ≤34 | 18 (21.4%) | 35 (15.0%) | 53 (16.7%) | 0.383 |
| 35–64 | 36 (42.9%) | 112 (47.9%) | 148 (46.5%) | |
| ≥65 | 30 (35.7%) | 87 (37.2%) | 117 (36.8%) | |
| Prior TB history | | | | |
| No | 72 (85.7%) | 194 (82.9%) | 266 (83.6%) | 0.551 |
| Yes | 12 (14.3%) | 40 (17.1%) | 52 (16.4%) | |
| Nationality | | | | |
| Korean | 80 (95.2%) | 220 (94.0%) | 300 (94.3%) | 0.678 |
| Foreign | 4 (4.8%) | 14 (6.0%) | 18 (5.7%) | |
| Comorbidities | | | | |
| No | 68 (81.0%) | 176 (75.2%) | 244 (76.7%) | 0.286 |
| Yes | 16 (19.0%) | 58 (24.8%) | 74 (23.3%) | |
| AFB smear result | | | | |
| Negative | 42 (50.0%) | 102 (43.6%) | 144 (45.3%) | |
| Positive | 42 (50.0%) | 132 (56.4%) | 174 (54.7%) | |
| INH resistance | | | | |
| High-resistant | 77 (91.7%) | 226 (96.6%) | 303 (95.3%) | 0.185 |
| Low-resistant or susceptible | 3 (3.6%) | 3 (1.3%) | 6 (1.9%) | |
| Not available | 4 (4.8%) | 5 (2.1%) | 9 (2.8%) | |
| Additional Fq use | | | | |
| No | 56 (66.7%) | 159 (67.9%) | 215 (67.6%) | 0.829 |
| Yes | 28 (33.3%) | 75 (32.1%) | 103 (32.4%) | |

H/INH, isoniazid; R, rifampicin; E, ethambutol; Z, pyrazinamide; Fq, fluoroquinolone; TB, tuberculosis.

Comorbidities included tuberculosis-related chronic diseases such as solid and hematologic malignancies, diabetes, transplantation, and human immunodeficiency virus infection.

Among 160 patients with positive treatment outcomes, the median duration of anti-TB treatment was 275.0 days (IQR = 249.8–290 days). Overall treatment duration of baseline regimen of 6-9REZ (median = 272.0, IQR = 234.8–286.3) was statistically shorter than that of 2REZ/7-10RE (median = 287.5, IQR = 274.8–365.0) ($P < 0.001$). However, overall treatment durations with and without additional Fq use (279.0 [263.0–293.0] vs. 274.0 [246.0–288.0]) were not statistically different ($P = 0.089$).

## Discussion

In this multicenter analysis of 318 patients with Hr-TB between 2011 and 2018, 6-9REZ was the most prescribed baseline regimen, and additional Fq was used in only one-third of the enrolled patients. Despite its retrospective design and small number of enrolled patients, the

**Table 2. Treatment outcomes among 318 enrolled pulmonary tuberculosis patients with isoniazid-resistant rifampicin-susceptible strain.**

| Treatment outcome | 2REZ/7-10RE | 6-9REZ | Total |
|---|---|---|---|
| | (n = 84) | (n = 234) | (n = 318) |
| Treatment success without recurrence | 42 (50.0%) | 118 (50.4%) | 160 (50.3%) |
| Unfavorable outcomes | 42 (50.0%) | 116 (49.6%) | 158 (49.7%) |
| Recurrence after treatment completion | 0 (0.0%) | 1 (0.4%) | 1 (0.3%) |
| Treatment failure | 21 (25.0%) | 94 (40.2%) | 115 (36.1%) |
| • Regimen changed due to adverse events | 6 (7.1%) | 36 (15.4%) | 42 (13.2%) |
| • Regimen strengthened due to worsening or not improving | 7 (8.3%) | 14 (6.0%) | 21 (6.6%) |
| • Extended length of treatment | 8 (9.5%) | 44 (18.8%) | 52 (16.4%) |
| Loss-to-follow-up | 15 (17.9%) | 5 (2.1%) | 20 (6.3%) |
| Death from any causes | 2 (2.4%) | 11 (4.7%) | 13 (4.1%) |
| • TB-associated death | 0 (0.0%) | 2 (0.9%) | 2 (0.6%) |
| • Death from non-TB related causes | 2 (2.4%) | 9 (3.8%) | 11 (3.5%) |
| Transferred to another hospital | 4 (4.8%) | 5 (2.1%) | 9 (2.8%) |

TB, tuberculosis.

**Table 3. Univariable logistic regression assessing factors associated with positive treatment outcomes.**

| | Total | Positive outcome | OR (95% CI) | *P* value |
|---|---|---|---|---|
| | n (column %) | n (row %) | | |
| Overall | 318 (100.0%) | 160 (50.3%) | | |
| Initial baseline regimen | | | | |
| 2REZ/7-10RE | 84 (26.4%) | 42 (50.0%) | | |
| 6-9REZ | 234 (73.6%) | 118 (50.4%) | 1.02 (0.62–1.68) | 0.946 |
| Additional Fq use | | | | |
| No | 215 (67.6%) | 103 (47.9%) | | |
| Yes | 103 (32.4%) | 57 (55.3%) | 1.35 (0.84–2.16) | 0.215 |
| Calendar year | | | | |
| 2011–2014 | 113 (35.5%) | 57 (50.4%) | | |
| 2015–2018 | 205 (64.5%) | 103 (50.2%) | 0.99 (0.63–1.57) | 0.973 |
| Age, years | | | | |
| ≤34 | 53 (16.7%) | 35 (66.0%) | | |
| 35–64 | 148 (46.5%) | 76 (51.4%) | 0.54 (0.28–1.04) | 0.067 |
| ≥65 | 117 (36.8%) | 49 (41.9%) | 0.37 (0.19–0.73) | 0.004 |
| Male | 195 (61.3%) | 94 (48.2%) | 0.80 (0.51–1.26) | 0.344 |
| Prior TB history | 52 (16.4%) | 26 (50.0%) | 0.98 (0.54–1.79) | 0.960 |
| Korean Nationality | 300 (94.3%) | 151 (50.3%) | 1.01 (0.39–2.62) | 0.978 |
| Comorbidities | 74 (23.3%) | 35 (47.3%) | 1.17 (0.70–1.97) | 0.554 |
| Positive AFB smear test result | 174 (54.7%) | 88 (50.6%) | 1.02 (0.66–1.59) | 0.919 |
| INH resistance | | | | |
| High-resistant | 303 (95.3%) | 156 (51.5%) | | |
| Low-resistant or susceptible | 6 (1.9%) | 2 (33.3%) | 0.47 (0.0–2.61) | 0.389 |
| Not available | 9 (2.8%) | 2 (22.2%) | 0.27 (0.06–1.32) | 0.105 |

OR, odds ratio; CI, confidence interval; H/INH, isoniazid; R, rifampicin; E, ethambutol; Z, pyrazinamide; Fq, fluoroquinolone; TB, tuberculosis; AFB, acid-fast bacilli. Comorbidities included tuberculosis-related chronic diseases such as solid and hematologic malignancies, diabetes, transplantation, and human immunodeficiency virus infection.

**Table 4. Multivariable logistic regression assessing factors associated with positive treatment outcomes.**

|  | model #1 | | model #2 | |
|---|---|---|---|---|
|  | aOR (95% CI) | *P* value | aOR (95% CI) | *P* value |
| Initial baseline regimen |  |  |  |  |
| 2REZ/7-10RE | 1.0 |  | 1.0 |  |
| 6-9REZ | 1.02 (0.62–1.68) | 0.934 | 1.08 (0.65–1.82) | 0.744 |
| Additional Fq use |  |  |  |  |
| No | 1.0 |  | 1.0 |  |
| Yes | 1.35 (0.84–2.16) | 0.215 | 1.41 (0.87–2.27) | 0.166 |
| Age, years |  |  |  |  |
| ≤34 |  |  | 1.0 |  |
| 35–64 |  |  | 0.58 (0.29–1.13) | 0.110 |
| ≥65 |  |  | 0.37 (0.19–0.74) | 0.005 |
| Male |  |  | 0.85 (0.52–1.37) | 0.498 |

H/INH, isoniazid; R, rifampicin; E, ethambutol; Z, pyrazinamide; Fq, fluoroquinolone; TB, tuberculosis.
Comorbidities included tuberculosis-related chronic diseases such as solid and hematologic malignancies, diabetes, transplantation, and human immunodeficiency virus infection.

strength of the study is the comparison of the effectiveness of four different regimens, namely: two initial baseline regimens (6-9REZ and 2REZ/7-10RE) with or without additional Fq. We found that compared to 2REZ/7-10RE, 6-9REZ had higher proportion of positive treatment outcome, however, its odds were not statistically significant. Although addition use of Fq was not associated with positive treatment outcomes in the whole cohort, subgroup analysis showed its additional use was beneficial in the 2REZ/7-10RE subgroup.

The WHO updated and published the treatment guidelines on the INH-resistant TB in 2018. Its evidence used to determine the composition and duration of regimens relied primarily on an analysis of individual patient database [12]. This research included only small number of enrolled patients was treated with the 2REZ/7-10RE regimen and could not fully evaluate its efficacy. However, shortening the PZA duration (ranging from one to three months) in a regimen that contains a Fq had high treatment success. It is well established that PZA is the most toxic among the present first-line anti-TB drugs, which raises concerns by experts about the longer duration of PZA use. This was reflected in the recent guidelines of the United States and Europe [13] that favor the shortening of the PZA duration when a later-generation Fq is included in the Hr-TB regimen of patients in whom there is anticipated or experienced toxicity of PZA or when the patient has a non-cavitary, lower burden of disease. In the United Kingdom, the National Institute for Health and Care Excellence (NICE) recommends a 9-month regimen of 2 months of RZE, followed by 7 months of RE [14]. A recent retrospective cohort study of 626 notified Hr-TB cases in London, which followed the NICE guideline, demonstrated that if the duration of (H)REZ treatment is long enough, a short PZA duration (median 2 months in the intensive phase) could be effective [9]. In addition, we demonstrated that when Fq was added to 2REZ/7-10RE, the odds of positive outcomes were significantly improved compared with no addition of Fq. Other potential advantages of adding Fq to 2REZ/7-10RE were reducing the extended treatment duration and minimizing changes in anti-TB drugs to strengthen the regimen. Our study results suggest that shortening PZA duration with additional Fq use might be a safe alternative for Hr-TB treatment. Further studies are necessary to support our result, which can guide physician to carefully select the Hr-TB patients who are eligible for short duration of PZA.

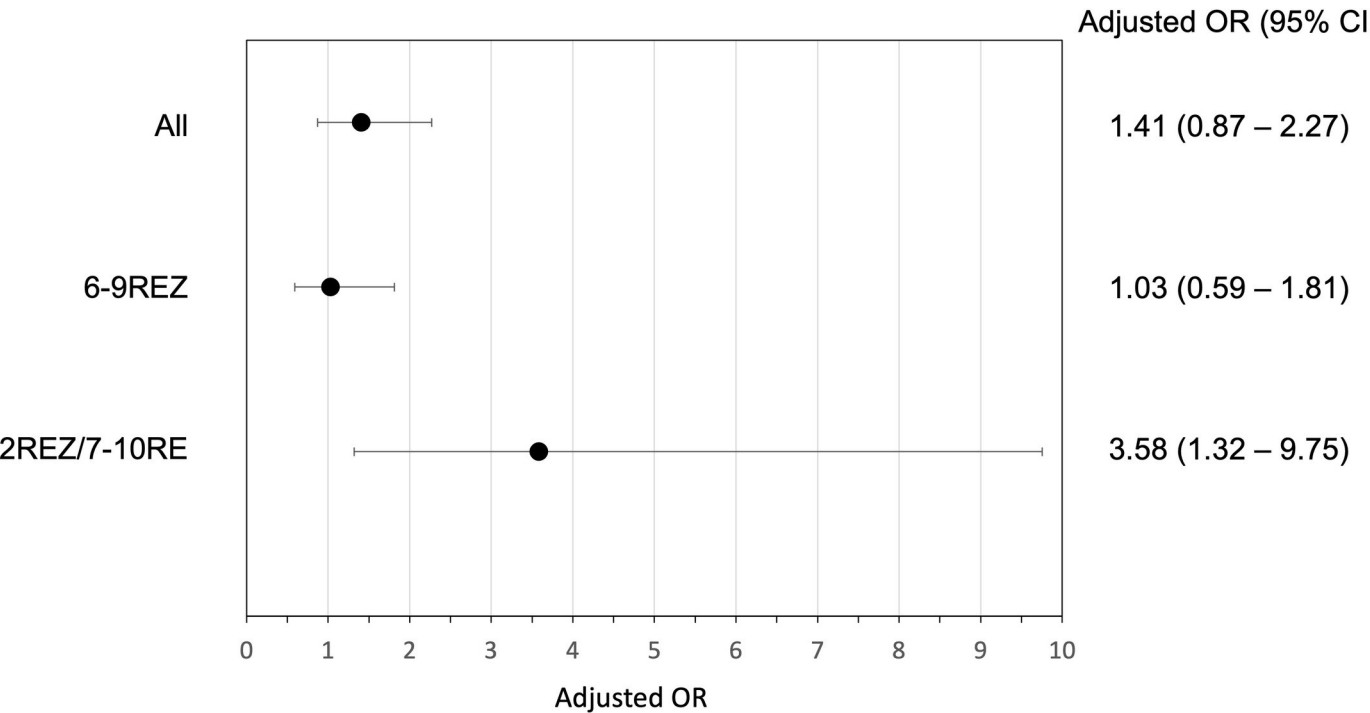

**Fig 2. Forest plot of odds ratios of positive treatment outcome by additional fluoroquinolone use, stratified by initial baseline regimens.** H, isoniazid; R, rifampicin; E, ethambutol; Z, pyrazinamide; Fq, fluoroquinolone; OR, odds ratio; CI, confidence interval.

Our results showed no benefits of adding Fq to the 6-9REZ, which could be ascribed to its long overall treatment duration of 9 months. It does not preclude its importance in the treatment of Hr-TB patients. Addition of Fq to 6 months or more of REZ was associated with a significantly greater success rate in the prior individual patient data meta-analysis [12]. As their mechanism of action is distinct, Fqs have become a mainstay of regimens used to treat drug-resistant TB. Its minimal inhibitory concentrations are low in both laboratory and clinical isolates and are well within usual serum concentrations with a generally good safety profile. More evidence is necessary to evaluate role of Fqs in order to optimize composition and duration of Hr-TB regimen.

The overall negative outcome rate was 49.7% in our study which is higher than that reported in previous studies. For our study, we adopted the WHO's new definitions of treatment outcomes and redefined "treatment failure" to capture patients who were treated for an extended period, which increased the proportion of negative outcomes. Because patients were enrolled between 2011 and 2018 before announcing the WHO's updated guideline on Hr-TB [5], Fq was not frequently added to the regimen in the current cohort. This phenomenon allowed clinicians to extend the treatment duration and add other second-line anti-TB drugs, which were the main causes of negative outcomes in our study. Among patients with positive outcomes, the overall treatment duration tended to be reduced when Fq was combined with the baseline regimen. Regarding adherence, treatment shortening is an important issue in patient-centered care.

INH resistance is classified into high- and low-level. Low-level resistance is mainly associated with mutations of *inhA* promoter genes [15]. We could not identify mutations of the *katG* gene and *inhA* promoter gene for all the enrolled participants, because molecular DST was not routinely performed in Korea during the study period. However, there were cases of

phenotypically INH-susceptible bacilli with *inhA* promoter gene mutations. Prior studies showed that including INH in the treatment regimens of low-level resistant strains may be acceptable [16]. In contrast, high-level resistance, which is mainly caused by *katG* mutations, were associated with an unfavorable treatment outcome [17]. We hypothesized that these differences in INH-resistant phenotypes could affect both physicians' prescription patterns and anti-TB treatment outcome. Our results revealed that low-level resistance and susceptibility to INH tended to have lower proportions of negative regimen-specific outcomes without statistical significance. Further larger studies are necessary to investigate the effect of low-level resistance.

This study has some limitations. First, the retrospective design could cause an information bias. Second, adverse drug reactions during anti-TB treatment could not be thoroughly analyzed because of a lack of systematic reports. Third, residual confounding might have occurred due to unmeasured differences in patient characteristics, such as nutritional status and genotypes of drug resistance. Fourth, the small number of patients with human immunodeficiency virus and diabetes limited generalizability to these important high-risk groups.

## Conclusion

In conclusion, compared to 6-9REZ, 2REZ/7-10RE was similarly effective and a safe alternative for patients with potential hepatotoxicity of Z. Additional use of Fq is beneficial for the initial baseline regimen of 2REZ/7-10RE. High-quality studies are necessary to optimize the composition and duration of Hr-TB regimens, particularly of Fqs and reducing the duration of pyrazinamide. Anti-TB treatment strategies are evolving from one-size-fits-all approach to individualized approach. We need stratifying strategy to identify suitable Hr-TB patients for each treatment regimen. Given the burden of Hr-TB worldwide, clinical trials to evaluate the efficacy, safety, and adherence of Hr-TB regimens are essential to provide stronger scientific evidence.

## Supporting information

**S1 File.**
(XLSX)

## Author Contributions

**Conceptualization:** Jinsoo Min, Hyung Woo Kim, Ju Sang Kim.

**Data curation:** Jinsoo Min, Ji Young Kang, Sung Kyoung Kim, Jin Woo Kim, Yong Hyun Kim, Hyoung Kyu Yoon, Sang Haak Lee, Ju Sang Kim.

**Formal analysis:** Jinsoo Min, Hyung Woo Kim, Ju Sang Kim.

**Funding acquisition:** Jinsoo Min.

**Writing – original draft:** Jinsoo Min, Hyung Woo Kim, Ju Sang Kim.

**Writing – review & editing:** Jinsoo Min, Hyung Woo Kim, Ji Young Kang, Sung Kyoung Kim, Jin Woo Kim, Yong Hyun Kim, Hyoung Kyu Yoon, Sang Haak Lee, Ju Sang Kim.

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
