## [Decision Letter · Decision Letter 0]

25 May 2022

PONE-D-22-11416Comparison of different regimens with or without fluoroquinolone  in isoniazid-resistant tuberculosis: a multicentre cohort studyPLOS ONE

Dear Dr. Ju Sang Kim,

Thank you for submitting your manuscript to PLOS ONE. After careful consideration, we feel that it has merit but does not fully meet PLOS ONE’s publication criteria as it currently stands. Therefore, we invite you to submit a revised version of the manuscript that addresses the points raised during the review process. We appreciate your study which is an interesting study. However, there are some points raised by the reviewers and need to be clarified including study criteria, treatment regimens and definition of outcomes. Please carefully respond to the reviewer comments and suggestions.

We look forward to receiving your revised manuscript.

Kind regards,

Vipa Thanachartwet, M.D.

Academic Editor

PLOS ONE

Journal Requirements:

Reviewers' comments:

Reviewer's Responses to Questions

**Comments to the Author**

1. Is the manuscript technically sound, and do the data support the conclusions?

Reviewer #1: Partly

Reviewer #2: Partly

2. Has the statistical analysis been performed appropriately and rigorously? 

Reviewer #1: Yes

Reviewer #2: No

3. Have the authors made all data underlying the findings in their manuscript fully available?

Reviewer #1: Yes

Reviewer #2: No

4. Is the manuscript presented in an intelligible fashion and written in standard English?

Reviewer #1: Yes

Reviewer #2: Yes

5. Review Comments to the Author

Reviewer #1: 1.The author did not mention about ethical approval by any hospital.

2.The author had to make clear how to count the starting time of study regimen because drug susceptibility was done by L-J media which required 2-3 months to get result. If the patients were started with 2HRZE, after 2 months treatment had to reduce to HR and then result of drug susceptibility test showed INH resistance. Treatment had to change for INH resistance. The treatment before changing to INH resistant regimen might affect outcomes of the study.

3.The author should explained the reason why selected duration of PZA use of 90 days to categorize regimen instead of 60 days.

4.In the negative outcomes criteria, one of them was "extended length of treatment" but the median time of treatment in 176 patients with positive outcomes was 274 days or 9 months and for 6RZE regimen was 272 days or 9 months. Both were longer than 6 months, will the author clarify this?

5.Should extrapulmonary TB include in the analysis because evaluation of outcomes for extrapulmonary TB was difficult?

6.Approximately 40% the study patients were smear negative, the author have to explain how to diagnose these patients and treatment was started by which reason?

7.Is there underlying reason to analyze INH resistant phenotype with treatment regimen and regimen outcomes?

8.The author should have a table to show details of negative outcomes in both 2RZE/7RE and 6RZEL because the author discussed that high negative outcomes were likely PZA toxicity.

Reviewer #2: Min and colleagues detail a comparative analysis of isoniazid-resistant regimens in light of recommendations from the World Health Organization. They find that fluoroquinolone use was not associated with significant improvements in outcomes, but that a 6-month regimen was associated with more negative outcomes than a 9 month regimen. This is an interesting and useful report, and I urge them to make their dataset freely accessible in a deidentified manner with their publication to permit its use in greater analyses of INHR-TB regimens. I have some other comments for the authors—some fundamental—before this can be considered acceptable for publication.

1. Calling outcomes “neutral” is uncommon and makes the generalizability of your findings difficult. I suggest you follow WHO outcome definitions and define cure or treatment completion without evidence of recurrence as “positive” and ALL other outcomes as “negative.” Death from any cause and loss to follow-up are unfavorable outcomes and should be treated as such. Please also ensure the timeframe for recurrence is consistent across the entire cohort (ie, likely only 1 year)—someone initiating treatment in 2011 has a longer timeframe for recurrence than someone initiating treatment in 2018 as it stands.

2. Is there any data to support the working definition of regimen classification? In other words, was there any independent validation from patient charts to ensure that when pyrazinamide was given for 90 days, indeed the patient received 6REZ? I especially bring this up given that those classified as 6REZ largely received 9 months of treatment (median = 272 days). As another point, how was “extended duration” of treatment defined, given that more than half of patients receiving 6REZ received >3 months of additional treatment?

3. How have outcomes changed over time in Korea? I would imagine they have significantly improved between 2011 and 2018. Would it make sense to try and control for time in your regression models to ensure that this temporal effect (that might be independent of regimen and associated with improvements in quality of care, for example) is adjusted for?

4. I am not sure I agree with the strength of language in the study conclusions. The WHO report was informed by this analysis: https://linkinghub.elsevier.com/retrieve/pii/S2213-2600(18)30078-X – which contained thousands of patients, while this is a comparatively smaller study. The direction of the point estimate remains consistent with the WHO analysis regarding fluoroquinolone use (beneficial). In addition, the finding that FQ use was not significantly beneficial in a 2REZ/7RE regimen is consistent with the findings of the above referenced analysis. It is not uncommon to find conflicting findings in the literature, but recommendations are developed on the entirety of the evidence base. Another issue is the differing outcome definitions used in this analysis and the one underpinning the WHO guidelines. I think the first paragraph of the discussion needs to be rephrased to recognize this and consistent definitions between studies need to be used to make statements about contrary findings. Since the authors do not provide the breakdown of neutral outcomes by regimens (only do so for negative), it is impossible for me to tell if this will bias the results. I strongly suggest the authors align their outcome definitions with WHO definitions.

5. “Native TB patients” – I assume this means people born in Korea. I would simply state this.

6. PLOS authors have the option to publish the peer review history of their article (what does this mean?). If published, this will include your full peer review and any attached files.

Reviewer #1: No

Reviewer #2: No

---

## [Author Response · Author response to Decision Letter 0]

7 Jul 2022

Editorial office:

1. Your ethics statement should only appear in the Methods section of your manuscript. If your ethics statement is written in any section besides the Methods, please move it to the Methods section and delete it from any other section. Please ensure that your ethics statement is included in your manuscript, as the ethics statement entered into the online submission form will not be published alongside your manuscript. 

[response] We moved our ethics statement from the Acknowledgement section to the Methods sections. Our ethics statement was in the page 7, line 169-172.

“This study was conducted in accordance with the principles of the Declaration of Helsinki. The Institutional Review Board (IRB) of the Catholic Medical Center, the Catholic University of Korea approved the study protocol (XC19REDE0040) and waived the need for informed consent because no patients were at risk.” (Page 7, line 169–172)

 

Reviewer #1: 

1.The author did not mention about ethical approval by any hospital.

[Response] The statement of the ethics approval was described in the Acknowledgment section was moved to the end of the Method section. The IRB of the Catholic Medical Center, the Catholic University of Korea approved the study protocol (XC19REDE0040). Our study is a multicenter retrospective study, which included eight university-affiliated hospitals of the Catholic University of Korea. The Catholic University of Korea operates its own IRB, which oversees any clinical studies at the eight hospitals.

“This study was conducted in accordance with the principles of the Declaration of Helsinki. The Institutional Review Board (IRB) of the Catholic Medical Center, the Catholic University of Korea approved the study protocol (XC19REDE0040) and waived the need for informed consent because no patients were at risk.” (Page 7, line 169–172)

2.The author had to make clear how to count the starting time of study regimen because drug susceptibility was done by L-J media which required 2-3 months to get result. If the patients were started with 2HRZE, after 2 months treatment had to reduce to HR and then result of drug susceptibility test showed INH resistance. Treatment had to change for INH resistance. The treatment before changing to INH resistant regimen might affect outcomes of the study.

[Response] Since INH resistance was detected both phenotypically and by molecular DSTs in our clinical settings in Korea, we could detect the INH resistance during the first and second months of anti-TB treatment. The starting time of the study regimen is the first day of anti-TB treatment, when the patient first took the anti-TB drugs. We counted all the prescribed doses of anti-TB drugs from the first day of anti-TB treatment for each patient. In order to clarify this, we added a sentence as follows.

“We counted all the prescribed doses of anti-TB drugs from the first day of anti-TB treatment for each patient.” (Page 5, line 114–115)

3.The author should explain the reason why selected duration of PZA use of 90 days to categorize regimen instead of 60 days.

[Response] Our study is based on a real-world database, which reflects the current clinical practice of anti-TB treatment in Korea. The Korean Academy of Tuberculosis and Lung Diseases regularly published the Korean TB guidelines from 1966, and it was last updated in 2020. Most Korean physicians prescribe the initial standard regimens of HREZ with 2-month use of PZA according to the guidelines. However, duration of PZA use may vary depending on the clinical judgements of the doctor. It could be extended due to accidental addition of anti-TB drug prescription. There were a few cases of prolonged PZA use of more than 60 days but less than 90 days without any comments or evidence of hepatotoxicity. In this case, we judged that the physician’s first intention was to administer the PZA for 60 days according to the guideline. Thus, we set the duration of PZA use to 90 days instead of 60 days. In order to clarify this, we added another sentence as follows. 

“Although the TB guidelines recommend initial prescription of standard regimens of HREZ with 2-month use of PZA, duration of PZA use may vary depending on clinical judgements and may be extended due to accidental addition of anti-TB drugs. There were a few cases of prolonged PZA use >60 days but <90 days without any comments or evidence of adverse events, so we judged that the physician’s first intention was to administer PZA for 60 days, according to the guideline. Therefore, we set the duration of PZA use to 90 days and classified it as 6REZ if PZA was administered for >90 days.” (Page 5, line 116–122)

4. In the negative outcomes criteria, one of them was "extended length of treatment" but the median time of treatment in 176 patients with positive outcomes was 274 days or 9 months and for 6RZE regimen was 272 days or 9 months. Both were longer than 6 months, will the author clarify this?

[Response] We thank the reviewer for the valuable comment. The 2008 WHO’s guideline of drug-resistant TB recommends prescription of REZ regimen between 6 and 9 months, which was adopted by the Korean TB guidelines. Therefore, we defined prescription of the 6REZ regimen for more than 10 months as “extended length of treatment.” 

In the United Kingdom, the 2016 National Institute for Health and Care Excellence recommends a 9-month regimen, comprising2 months of REZ followed by 7 months of RE. This can be extended to 12 months if disease is extensive. We adopted this guideline and defined a 2REZ/7RE regimen of more than 13 months as “extended length of treatment”.

Because of these pre-specified definitions of extended length of treatment, the median time of anti-TB treatment among those with positive outcomes was longer than 6 months. 

“The 2008 WHO’s guideline of drug-resistant TB recommends prescription of the REZ regimen for 6–9 months; we defined a 6REZ regimen of more than 10 months as “extended length of treatment.” In the United Kingdom, the 2016 National Institute for Health and Care Excellence recommends a 9-month regimen comprising2 months of REZ, followed by 7 months of RE. This can be extended to 12 months if disease is extensive. We adopted this guideline and defined a 2REZ/7RE regimen of more than 13 months as “extended length of treatment.”” (Page 5, line 104–110)

5.Should extrapulmonary TB include in the analysis because evaluation of outcomes for extrapulmonary TB was difficult?

[Response] Thank you for your insightful comment. We agree that extrapulmonary TB could affect the treatment outcomes. Hence, we excluded 38 patients with extrapulmonary TB. We modified the inclusion and exclusion criteria in the article as well as the flowchart of study participant enrollment (Figure 1).

“We included notified patients with pulmonary TB aged ≥15 years between January 2011 and December 2018 in the study cohort.” (Page 4, line 76–77)

“Exclusion criteria were as follows: … (3) patients with only extrapulmonary TB.” (Page 4, line 83)

“After excluding 49 patients who had their initial treatment regimen changed within the first 2 months and 38 patients with extrapulmonary tuberculosis, 318 patients were finally enrolled for analysis.” (Page 7, line 176–178)

6.Approximately 40% the study patients were smear negative, the author have to explain how to diagnose these patients and treatment was started by which reason?

[Response] One of the inclusion criteria is phenotypically or genotypically confirmed INH resistance. All the patients included in the study had a positive result for the AFB culture test and thus, their diagnosis was confirmed. In order to clarify this, we added another inclusion criterion as follows.

“(1) patients who had a positive acid-fast bacillus culture test result” (Page 4, line 79)

7.Is there underlying reason to analyze INH resistant phenotype with treatment regimen and regimen outcomes?

[Response] Thank you again for your valuable comment. INH resistance is classified into high- and low-level resistance. Low-level resistance is mainly associated with inhA promoter mutation. Some evidence suggests that including INH in the treatment regimens of low-level resistant strains may be acceptable. In addition, we identified discrepant results between phenotypes and molecular DSTs. For example, there were some cases of phenotypically INH-susceptible MTBc with inhA gene mutations. We thought that these differences within an INH-resistant phenotype could affect both physicians’ prescription patterns and anti-TB treatment outcome. 

Our results revealed that, compared to high-level resistance, low-level resistance or susceptibility of INH had lower proportions of negative regimen-specific outcomes without statistical significance. Further larger studies are necessary to investigate the roles of low-level INH resistance. We added an additional paragraph regarding this issue in the discussion section.

“INH resistance is classified into high- and low-level. Low-level resistance is mainly associated with mutations of inhA promoter genes. We could not identify mutations of the katG gene and inhA promoter gene for all the enrolled participants, because molecular DST was not routinely performed in Korea during the study period. However, there were cases of phenotypically INH-susceptible bacilli with inhA promoter gene mutations. Prior studies showed that including INH in the treatment regimens of low-level resistant strains may be acceptable. In contrast, high-level resistance, which is mainly caused by katG mutations, were associated with an unfavorable treatment outcome. We hypothesized that these differences in INH-resistant phenotypes could affect both physicians’ prescription patterns and anti-TB treatment outcome. Our results revealed that low-level resistance and susceptibility to INH tended to have lower proportions of negative regimen-specific outcomes without statistical significance. Further larger studies are necessary to investigate roles of low-level resistance.” (page 14, line 294–305)

8.The author should have a table to show details of negative outcomes in both 2RZE/7RE and 6RZEL because the author discussed that high negative outcomes were likely PZA toxicity.

[Response] We modified Table 2 to show details of negative outcomes and compare them between the 2RZE/7RE and 6REZ. It is clear that the most common causes of negative outcomes were loss-to-follow-up for the 2REZ/7RE group and extended length of treatment for 6REZ. Regimen changes due to adverse events were higher in the 6REZ group, compared to the 2REZ/7RE. Reviewer 2 suggested following the WHO outcome definitions and define cure or treatment completion without evidence of recurrence as “positive” and all other outcomes as “negative.” Thus, we further modified the Table 2 as follows. 

Treatment outcome 2REZ/7RE 6REZ Total

 (n = 84) (n = 234) (n = 318)

Treatment success without recurrence 42 (50.0%) 118 (50.4%) 160 (50.3%)

Unfavorable outcomes 42 (50.0%) 116 (49.6%) 158 (49.7%)

Recurrence after treatment completion 0 (0.0%) 1 (0.4%) 1 (0.3%)

Treatment failure 21 (25.0%) 94 (40.2%) 115 (36.1%)

- Regimen changed due to adverse events 6 (7.1%) 36 (15.4%) 42 (13.2%)

- Regimen strengthened due to worsening or not improving 7 (8.3%) 14 (6.0%) 21 (6.6%)

- Extended length of treatment 8 (9.5%) 44 (18.8%) 52 (16.4%)

Loss-to-follow-up 15 (17.9%) 5 (2.1%) 20 (6.3%)

Death from any causes 2 (2.4%) 11 (4.7%) 13 (4.1%)

- TB-associated death 0 (0.0%) 2 (0.9%) 2 (0.6%)

- Death from non-TB related causes 2 (2.4%) 9 (3.8%) 11 (3.5%)

Transferred to another hospital 4 (4.8%) 5 (2.1%) 9 (2.8%)

(Page 9) 

Reviewer #2: Min and colleagues detail a comparative analysis of isoniazid-resistant regimens in light of recommendations from the World Health Organization. They find that fluoroquinolone use was not associated with significant improvements in outcomes, but that a 6-month regimen was associated with more negative outcomes than a 9 month regimen. This is an interesting and useful report, and I urge them to make their dataset freely accessible in a deidentified manner with their publication to permit its use in greater analyses of INHR-TB regimens. I have some other comments for the authors—some fundamental—before this can be considered acceptable for publication.

[Response] We prepared the dataset in a de-identified manner and uploaded them as a supplementary material.

1. Calling outcomes “neutral” is uncommon and makes the generalizability of your findings difficult. I suggest you follow WHO outcome definitions and define cure or treatment completion without evidence of recurrence as “positive” and ALL other outcomes as “negative.” Death from any cause and loss to follow-up are unfavorable outcomes and should be treated as such. Please also ensure the timeframe for recurrence is consistent across the entire cohort (i.e., likely only 1 year)—someone initiating treatment in 2011 has a longer timeframe for recurrence than someone initiating treatment in 2018 as it stands.

[Response] We thank the reviewer for this critical comment. In agreement with this, we modified our outcome definitions according to those of the WHO. We re-categorized ‘death from any causes’ and ‘loss-to-follow-up’ into the unfavorable outcomes. 

In 2020, the WHO convened a consultation meeting to update the definitions of drug-resistant TB treatment outcomes (Meeting report of the WHO expert consultation on drug-resistant tuberculosis treatment outcome definitions, 17–19 November 2020). This consultation led the WHO to propose new definitions of treatment outcomes for both DS-TB and DR-TB and defined a failed treatment as “a patient whose treatment regimen needed to be terminated or permanently changed to a new regimen or treatment strategy.” “No response to treatment” is one of the most important reasons for treatment failure. Therefore, when treatment duration is extended due to lack of clinical response, we defined it as “treatment failure.” 

We confirm that all the patients were followed up for at least one year after treatment completion in order to identify TB recurrence.

We renamed the subheading and modified its contents as follows. 

“Treatment outcomes

Treatment outcomes were defined according to the WHO’s definition. “Treatment success” was defined as treatment completed as initially prescribed once INH-resistance was known, without extending duration of specified regimens. “Positive treatment outcome” was defined as success of treatment without recurrence within the 1-year post-treatment follow-up period. “Unfavorable outcome” was defined as a composite outcome that includes death, treatment failure, loss-to-follow-up, transfer-out, and recurrence.

Recently, the WHO convened a consultation meeting to update treatment outcome definitions (8), in which they proposed a new definition of “treatment failure”—when a treatment regimen is terminated or permanently changed to a new treatment or treatment strategy. Because one of our study objectives was to evaluate the efficacy and toxicity of treatment regimens for Hr-TB, we adopted concept of regimen-specific outcomes (9, 10) and defined treatment failure as follows: (1) regimen changed due to adverse events; (2) regimen strengthened due to worsening or not improving; and (3) extended length of treatment due to lack of clinical response. 

“No response to treatment” is one of the most important reasons for treatment failure. Therefore, when treatment duration was extended due to lack of clinical response, we defined it as a treatment failure according to the revised WHO’s outcome definitions. The 2008 WHO’s guideline of drug-resistant TB recommends the prescription of the REZ regimen for 6–9 months; we defined a 6REZ regimen of more than 10 months as “extended length of treatment.” In the United Kingdom, the 2016 National Institute for Health and Care Excellence recommends a 9-month regimen comprising 2 months of REZ, followed by 7 months of RE. This can be extended to 12 months if disease is extensive. We adopted this guideline and considered those prescribed a 2REZ/7RE regimen of more than 13 months as having “extended length of treatment.” (page 4–5, line 87–110)

2. Is there any data to support the working definition of regimen classification? In other words, was there any independent validation from patient charts to ensure that when pyrazinamide was given for 90 days, indeed the patient received 6REZ? I especially bring this up given that those classified as 6REZ largely received 9 months of treatment (median = 272 days). As another point, how was “extended duration” of treatment defined, given that more than half of patients receiving 6REZ received >3 months of additional treatment?

[response] Our study is based on a real-world database, which reflects the current clinical practice of anti-TB treatment in Korea. The Korean Academy of Tuberculosis and Lung Diseases regularly published the Korean TB guideline from 1966, and it was last updated in 2020. Most Korean physicians initially prescribe the standard regimens of HREZ with 2-month use of PZA, according to guidelines. However, duration of PZA use may vary depending on the clinical judgements of the doctor. It could be extended due to accidental addition of anti-TB drug prescription. There were a few cases of prolonged PZA use >60 days but <90 days without any comments or evidence of hepatotoxicity. In this case, we inferred that the physicians’ first intention was to administer PZA for 60 days according to the guideline. Thus, we set the duration of PZA use to 90 days instead of 60 days. In order to clarify this, we added another sentence as follows. 

“Although the TB guidelines recommend initial prescription of standard regimens of HREZ with 2-month use of PZA, duration of PZA use may vary depending on clinical judgements and may be extended due to accidental addition of anti-TB drugs. There were a few cases of prolonged PZA use of >60 days but <90 days without any comments or evidence of adverse events, so we inferred that the physician’s first intention was to administer PZA for 60 days, according to the guideline. Therefore, we set the duration of PZA use to 90 days and classified it as 6REZ if PZA was administered for >90 days.” (Page 5, line 116–124)

The 2008 WHO’s guideline of drug-resistant TB recommends prescription of the REZ regimen for 6–9 months, which was adopted by the Korean TB guidelines. Therefore, we defined a 6REZ regimen of more than 10 months as “extended length of treatment”. 

In the United Kingdom, the 2016 National Institute for Health and Care Excellence recommends a 9-month regimen comprising 2 months of REZ followed by 7 months of RE. This can be extended to 12 months if disease is extensive. We adopted this guideline and defined a 2REZ/7RE regimen of more than 13 months as “extended length of treatment.”

Because of these pre-specified definitions of extended length of treatment, the median time of anti-TB treatment among those with positive outcomes was longer than 6 months. 

“The 2008 WHO’s guideline of drug-resistant TB recommends prescription of the REZ regimen for 6–9 months; we defined a 6REZ regimen of more than 10 months as “extended length of treatment.” In the United Kingdom, the 2016 National Institute for Health and Care Excellence recommends a 9-month regimen comprising 2 months of REZ followed by 7 months of RE. This can be extended to 12 months if disease is extensive. We adopted this guideline and defined a 2REZ/7RE regimen of more than 13 months as “extended length of treatment.”” (Page 5, line 104–110)

3. How have outcomes changed over time in Korea? I would imagine they have significantly improved between 2011 and 2018. Would it make sense to try and control for time in your regression models to ensure that this temporal effect (that might be independent of regimen and associated with improvements in quality of care, for example) is adjusted for?

[Response] We made “calendar year” an independent variable (i.e., “2011–2014” vs. “2015–2018”). We compared treatment outcomes between the two periods, which showed that they were similar.

We included “calendar year” as a baseline characteristic in Table 1 (page 8) and Table 3 (page 10) as follows. 

 Total Positive outcome OR (95% CI) P value

 n (column %) n (row %) 

Overall 318 (100.0%) 160 (50.3%) 

Calendar year 

2011–2014 113 (35.5%) 57 (50.4%) 

2015–2018 205 (64.5%) 103 (50.2%) 0.99 (0.63–1.57) 0.973

4. I am not sure I agree with the strength of language in the study conclusions. The WHO report was informed by this analysis: https://linkinghub.elsevier.com/retrieve/pii/S2213-2600(18)30078-X – which contained thousands of patients, while this is a comparatively smaller study. The direction of the point estimate remains consistent with the WHO analysis regarding fluoroquinolone use (beneficial). In addition, the finding that FQ use was not significantly beneficial in a 2REZ/7RE regimen is consistent with the findings of the above referenced analysis. It is not uncommon to find conflicting findings in the literature, but recommendations are developed on the entirety of the evidence base. Another issue is the differing outcome definitions used in this analysis and the one underpinning the WHO guidelines. I think the first paragraph of the discussion needs to be rephrased to recognize this and consistent definitions between studies need to be used to make statements about contrary findings. Since the authors do not provide the breakdown of neutral outcomes by regimens (only do so for negative), it is impossible for me to tell if this will bias the results. I strongly suggest the authors align their outcome definitions with WHO definitions.

[Response] We thank the reviewer for this valuable comment. We redefined our primary outcome based on the WHO definition. Because of this change, we modified the Results, Discussion, and Conclusion sections accordingly. We also modified the abstract. Major findings of our study are: (1) outcomes of 2REZ/7RE are similar to that of 6REZ; and (2) it is beneficial to add Fq to initial baseline regimen of 2REZ/7RE. We think that shortening of PZA duration with additional Fq use could be a safe alternative for Hr-TB treatment. 

5. “Native TB patients” – I assume this means people born in Korea. I would simply state this.

[Response] Thank you for the comment. In order to clarify this, we changed the name of the aforementioned variable to “Korean nationality.” (Table 3, page 10)

---

## [Decision Letter · Decision Letter 1]

1 Aug 2022

PONE-D-22-11416R1Comparison of different regimens with or without fluoroquinolone in isoniazid-resistant tuberculosis: A multicenter cohort studyPLOS ONE

Dear Dr. Ju Sang Kim,

Thank you for submitting your manuscript to PLOS ONE. After careful consideration, we feel that it has merit but does not fully meet PLOS ONE’s publication criteria as it currently stands. Therefore, we invite you to submit a revised version of the manuscript that addresses the points raised during the review process. We appreciate your efforts for the study and the authors have made a careful revision to the manuscript. However, there are some minor points raised by the reviewer’s. Please carefully respond to the reviewer comments and suggestions.

Please submit your revised manuscript by  Sep 15 2022 11:59PM. If you will need more time than this to complete your revisions, please reply to this message or contact the journal office at plosone@plos.org. Please include the following items when submitting your revised manuscript:A rebuttal letter that responds to each point raised by the academic editor and reviewer(s). You should upload this letter as a separate file labeled 'Response to Reviewers'.A marked-up copy of your manuscript that highlights changes made to the original version. You should upload this as a separate file labeled 'Revised Manuscript with Track Changes'.An unmarked version of your revised paper without tracked changes. You should upload this as a separate file labeled 'Manuscript'.If applicable, we recommend that you deposit your laboratory protocols in protocols.io to enhance the reproducibility of your results. Protocols.io assigns your protocol its own identifier (DOI) so that it can be cited independently in the future. For instructions see: https://journals.plos.org/plosone/s/submission-guidelines#loc-laboratory-protocols. Additionally, PLOS ONE offers an option for publishing peer-reviewed Lab Protocol articles, which describe protocols hosted on protocols.io. Read more information on sharing protocols at https://plos.org/protocols?utm_medium=editorial-email&utm_source=authorletters&utm_campaign=protocols.

We look forward to receiving your revised manuscript.

Kind regards,

Vipa Thanachartwet, M.D.

Academic Editor

PLOS ONE

Journal Requirements:

Reviewers' comments:

Reviewer's Responses to Questions

**Comments to the Author**

1. If the authors have adequately addressed your comments raised in a previous round of review and you feel that this manuscript is now acceptable for publication, you may indicate that here to bypass the “Comments to the Author” section, enter your conflict of interest statement in the “Confidential to Editor” section, and submit your "Accept" recommendation.

Reviewer #2: (No Response)

2. Is the manuscript technically sound, and do the data support the conclusions?

Reviewer #2: Yes

3. Has the statistical analysis been performed appropriately and rigorously? 

Reviewer #2: Yes

4. Have the authors made all data underlying the findings in their manuscript fully available?

Reviewer #2: Yes

5. Is the manuscript presented in an intelligible fashion and written in standard English?

Reviewer #2: Yes

6. Review Comments to the Author

Reviewer #2: I thank the authors for the numerous edits to the manuscript, which I think is much improved. I have 1 request, based on the response to the comments on regimen duration. The 6REZ regimen was considered extended if >10 months and the 2/7 regimen considered extended if >13 months. I would rename these regimens: 6-9REZ and 2REZ/7-10RE -- just as we do with standardized short MDRTB regimens.

7. PLOS authors have the option to publish the peer review history of their article (what does this mean?). If published, this will include your full peer review and any attached files.

Reviewer #2: No

---

## [Author Response · Author response to Decision Letter 1]

1 Aug 2022

Reviewer #2: I thank the authors for the numerous edits to the manuscript, which I think is much improved. I have 1 request, based on the response to the comments on regimen duration. The 6REZ regimen was considered extended if >10 months and the 2/7 regimen considered extended if >13 months. I would rename these regimens: 6-9REZ and 2REZ/7-10RE -- just as we do with standardized short MDRTB regimens.

[response] I truly appreciate another opportunity to revise the manuscript. I understood the valuable comment and renamed the regimens in the manuscript and figures, accordingly. I changed 6REZ to 6-9REZ. I changed 2REZ/7RE to 2REZ/7-10RE.

---

## [Decision Letter · Decision Letter 2]

3 Aug 2022

PONE-D-22-11416R2Comparison of different regimens with or without fluoroquinolone in isoniazid-resistant tuberculosis: A multicenter cohort studyPLOS ONE

Dear Dr. Takeshi Horii,

Thank you for submitting your manuscript to PLOS ONE. After careful consideration, we feel that it has merit but does not fully meet PLOS ONE’s publication criteria as it currently stands. Therefore, we invite you to submit a revised version of the manuscript that addresses the points raised during the review process. The authors corrected nearly all comments raised by the reviewer’s. However, there are some minor points need to be corrected. Please correct as suggested. 

We look forward to receiving your revised manuscript.

Kind regards,

Vipa Thanachartwet, M.D.

Academic Editor

PLOS ONE

Journal Requirements:

Additional Editor Comments:

Regarding treatment outcome (lines 104-110, page 5) in materials and methods, the abstract (lines 35-38, page 2) should be corrected accordingly. Therefore, “Baseline regimens were classified into two groups, namely 6-month rifampicin, ethambutol, and pyrazinamide (6-9REZ) and a combination regimen of 2-month rifampicin, ethambutol, pyrazinamide and 7-month rifampicin and ethambutol (2REZ/7-10RE).” in the abstract should be corrected to “Baseline regimens were classified into two groups, namely 6-9 month rifampicin, ethambutol, and pyrazinamide (6-9REZ) and a combination regimen of 2-month rifampicin, ethambutol, pyrazinamide and 7-10 month rifampicin and ethambutol (2REZ/7-10RE).”.

Reviewers' comments:

Reviewer's Responses to Questions

**Comments to the Author**

1. If the authors have adequately addressed your comments raised in a previous round of review and you feel that this manuscript is now acceptable for publication, you may indicate that here to bypass the “Comments to the Author” section, enter your conflict of interest statement in the “Confidential to Editor” section, and submit your "Accept" recommendation.

Reviewer #2: All comments have been addressed

2. Is the manuscript technically sound, and do the data support the conclusions?

Reviewer #2: (No Response)

3. Has the statistical analysis been performed appropriately and rigorously? 

Reviewer #2: (No Response)

4. Have the authors made all data underlying the findings in their manuscript fully available?

Reviewer #2: (No Response)

5. Is the manuscript presented in an intelligible fashion and written in standard English?

Reviewer #2: (No Response)

6. Review Comments to the Author

Reviewer #2: (No Response)

7. PLOS authors have the option to publish the peer review history of their article (what does this mean?). If published, this will include your full peer review and any attached files.

Reviewer #2: No

---

## [Author Response · Author response to Decision Letter 2]

3 Aug 2022

Additional Editor Comments:

Regarding treatment outcome (lines 104-110, page 5) in materials and methods, the abstract (lines 35-38, page 2) should be corrected accordingly. Therefore, “Baseline regimens were classified into two groups, namely 6-month rifampicin, ethambutol, and pyrazinamide (6-9REZ) and a combination regimen of 2-month rifampicin, ethambutol, pyrazinamide and 7-month rifampicin and ethambutol (2REZ/7-10RE).” in the abstract should be corrected to “Baseline regimens were classified into two groups, namely 6-9 month rifampicin, ethambutol, and pyrazinamide (6-9REZ) and a combination regimen of 2-month rifampicin, ethambutol, pyrazinamide and 7-10 month rifampicin and ethambutol (2REZ/7-10RE).”.

[response] Thank you for your comment. We changed it as follows.

“Baseline regimens were classified into two groups, namely 6–9-month rifampicin, ethambutol, and pyrazinamide (6-9REZ) and a combination regimen of 2-month rifampicin, ethambutol, pyrazinamide and 7–10-month rifampicin and ethambutol (2REZ/7-10RE).”

---

## [Editor Report · Decision Letter 3]

5 Aug 2022

Comparison of different regimens with or without fluoroquinolone in isoniazid-resistant tuberculosis: A multicenter cohort study

PONE-D-22-11416R3

Dear Dr. Ju Sang Kim,

We’re pleased to inform you that your manuscript has been judged scientifically suitable for publication and will be formally accepted for publication once it meets all outstanding technical requirements.

Kind regards,

Vipa Thanachartwet, M.D.

Academic Editor

PLOS ONE

Additional Editor Comments (optional):

All issues were revised according to the comments and suggestions.
---

## [Editor Report · Acceptance letter]

10 Aug 2022

PONE-D-22-11416R3 

Comparison of different regimens with or without fluoroquinolone in isoniazid-resistant tuberculosis: A multicenter cohort study 

Dear Dr. Kim:

I'm pleased to inform you that your manuscript has been deemed suitable for publication in PLOS ONE. Congratulations! Your manuscript is now with our production department. 

Kind regards, 

on behalf of

Associate Professor Vipa Thanachartwet 

Academic Editor

PLOS ONE